# Effects of Social Networks in Promoting Young Adults’ Physical Activity among Different Sociodemographic Groups

**DOI:** 10.3390/bs12090345

**Published:** 2022-09-19

**Authors:** Ting Du, Yingru Li

**Affiliations:** 1Department of Humanities and Social Development, Northwest A&F University, Shaanxi 712100, China; 2Department of Sociology, University of Central Florida, Orlando, FL 32816, USA

**Keywords:** social network, physical activity, sociodemographic groups

## Abstract

Introduction: Physical inactivity has become a public health issue as it can trigger many chronic diseases. Studies have found that an individual’s social networks (SNs) influence their engagement in physical activity (PA). However, it remains unclear how the influence varies between different sociodemographic groups. This study examined the associations between the SN structures and the PA of young adults across sociodemographic groups. Method: Data on 14,595 young adults aged 24–32 were obtained from Wave IV of the National Longitudinal Study of Adolescent to Adult Health. Latent class analysis was conducted to identify heterogeneous subgroups of respondents with respect to their engagement in PA and SN structures were measured in terms of the relationship quality, contact frequency, network size, and spatial accessibility. Logistic regression and chi-square tests were used to further test the relationships between PA and SNs as well as the sociodemographic factors. Results: SNs were found to have a significant influence on PA. Relationship quality was identified as the most important feature of an individual’s SN, followed by network size, contact frequency, and accessibility. The effects of SNs on PA varied with the individuals’ gender, ethnicity, income, and educational attainment. For example, male participants tended to be physically active if they had frequent connections with their neighbors, while the engagement in PA of the female participants was strongly related to the frequency of their contact with their children. Conclusions: This research has important implications for practitioners as it suggests that PA intervention programs should be designed to target specific population groups.

## 1. Introduction

Physical inactivity is the fourth leading risk factor for mortality worldwide, contributing to 6–10% of deaths from non-communicable diseases, according to The Lancet [1,2,3]. Individuals are defined as physically active if they engage in at least 75 to 150 min of vigorous-intensity aerobic physical activity (PA) per week or if they engage in weight training or muscle-strengthening activities that work all major muscle groups on two or more days per week [4]. However, around 40% of adults in the USA and one in four adults worldwide are considered to be physically inactive [5]. Understanding the mechanisms underlying an individual’s engagement in PA is crucial to enable policy makers and health professionals to implement relevant policies and interventions to support and promote people’s engagement in PA.

Having a supportive social network (SN) has been identified as an important factor promoting PA [6,7]. “SN” refers to the web of social interactions and relationships surrounding an individual [8,9,10]. SNs significantly influence an individual’s behavior by providing them with social support and access to social resources [11]. The individual’s different social roles—as parents, spouses, neighbors, peers, colleagues, etc.—can be defined in a diverse network that conveys shared norms [11]. Individuals who have supportive SNs tend to have better access to either tangible or intangible social resources through supportive ties in their SNs, which further facilitate their engagement in PA [10,12]. Those who lack social ties or have limited connections to others in their networks tend to receive less support and are more likely to suffer from psychological problems such as stress, anxiety, or panic, which may lead to unexpected negative behavioral changes such as reduced exercise [11,13,14].

The SN structure can be measured in terms of quality (supportive or inhibitory relationships), frequency (communication or contact frequency between two nodes), size (number of members in an SN), and accessibility (degree of difficulty of reaching other members) [7,15]. Studies have found that high-quality networks, especially with immediate family members, spouses/partners, and colleagues, are associated with increased PA [16,17,18]. Network size has also been shown to be associated with an individual’s health behaviors [18,19,20]. For instance, large friendship networks promote the PA of adults [20]. Contact frequency, another important measure of network structure, has been found to be conducive to increased leisure time PA [18,21]. Spatial accessibility is another vital feature of an SN, capturing the impact of distance on contact frequency [22]. Geographical proximity increases the likelihood of social interactions [22], which further impacts the individual’s activity levels [23]. However, the findings regarding the association between SNs and PA are not consistent. For example, some scholars have found that individuals with few or no connections with others such as neighbors are more physically active than individuals who have some or many social connections [20]. However, other scholars have claimed that exercise habits can be undermined by intimate relationships with spouses/partners, family, friends, and coworkers [24]. A possible reason for these inconsistent findings is that each of these studies used a distinct aspect of network structure to capture the effects of SNs, which may have undermined their assessment of the effects of SNs on PA or even created reverse associations [25,26,27].

The effects of SNs on PA may vary between sociodemographic groups [28,29,30]. Individuals with higher social status are more likely to have supportive SNs and adequate resources, which promote their engagement in PA [29,30,31]. For example, wealthy and well-educated people commonly possess ample social and material resources (e.g., gym memberships, access to open space) that facilitate their engagement in PA [32]. Gender differences have also been found to affect the influence of SNs on PA. Men are more likely to be influenced by their friendship networks than women [33]. Regarding different racial groups, Hays and Mindel [34] found that Black people tended to receive more social support from their families than White people, while Griffin and his colleagues [35] observed that White people had more supportive friendship networks and more frequent connections with friends than other racial groups.

Although SNs play an important role in people’s lives, it remains unclear how the effects of SNs on PA vary across sociodemographic groups. To fill this research gap, this study aimed to explore the effects of different aspects of SNs on an individual’s PA engagement and examine how these effects varied between sociodemographic groups among young adults. We hypothesized that the individual’s SNs, measured in terms of the contact frequency, quality, size, and accessibility, have a significant influence on their engagement in PA and that the effects of SNs on PA vary with the individual’s sociodemographic characteristics such as gender, race, income, and educational attainment.

## 2. Methods

### 2.1. Data

The data required for this study were obtained from Wave IV of the National Longitudinal Study of Adolescent to Adult Health (Add Health) in the United States. One thousand one hundred and six (*n* = 1106) participants were omitted from the analysis due to missing data. All other participants (*n* = 14,595) with complete data were included in the analysis (Table 1). When the survey was conducted, the participants were in early adulthood, aged 24 to 32. Among them, 53.55% (*n* = 7816) were physically active. In terms of sex, the participants included 53.56% of women and 46.44% men, and 70.19% White and 29.81% racial/ethnic minorities. Regarding the annual household income, 19.78% of the participants came from low-income families (≤$30,000), 64.93% were from mid-income families ($30,000–$100,000), and 15.29% were from high-income families (≥$100,000) [36]. The participants’ education levels also varied. Only 7.64% of the participants had not received a high school education or equivalent, while 60.51% had completed high school or equivalent, and 31.84% had received a college education or above [37].

### 2.2. Methods

Latent class analysis (LCA) was conducted to examine the associations between the young adults’ SNs and their engagement in PA. LCA is a person-centered statistical method that assigns participants with common characteristics to homogeneous classes, while individuals are heterogeneous between classes [38]. The goal of LCA is to distinguish potential classes in a sample and classify participants based on the latent relationships between behavioral variables [39]. This approach outperforms traditional variable-centered segmentation methods (e.g., cluster analysis) because it permits statistical inference based on a likelihood model [38]. LCA was run with the Mplus version 7.0 software program (Mplus, Los Angeles, CA). The latent class indicators were dichotomously coded and included the following variables.

***PA level.*** An individual’s PA level was assessed using seven questions with the same question format: “In the last seven days, how many times have you [performed a specified type of PA]?” The targeted activities included both individual and team sports such as football, soccer, wrestling, and swimming, which are considered to be moderately intense to vigorous aerobic activities [40,41]. The total duration of engagement in these activities was calculated to reflect each individual’s PA level. A participant was considered to have a high PA level if they reported five or more instances of these activities; otherwise, their PA level was low [40,41].

***SN factors.*** SN structures were measured by the frequency of each individual’s contact with their father, mother, child(ren), and neighbors; the quality of their relationships with their father, mother, child(ren), spouse/partner, and colleagues; the size of their network of friends and siblings; and their access to father, mother, child(ren), and spouse/partner. Each network structure was measured with one or more relationship type, depending on the characteristics of the relationship type and the availability of the survey data.

**Contact frequency** was measured by an individual’s frequency of interaction with their father, mother, child(ren), and neighbors. Frequency of contact with parents was reflected by the participants’ responses to two questions: “How often do you and your mother/father see each other?” and “How often do you and your mother/father contact each other by telephone, mail, or emails?”. The responses were provided on a 6-point scale from 0 to 5, indicating answers from “never” to “almost every day” in sequence. Answers of “almost every day”, “once or twice a day”, and “once or twice a month” were coded as 1; other answers were coded as 0 [42]. A summed value of 1 or 2 was recoded as 1 to represent frequent contact with parents, and otherwise 0, reflecting infrequent contact. Frequency of contact with child(ren) and frequency of contact with neighbors, respectively, were coded as dichotomous variables, according to the survey questions and relevant literature [42].

**Quality** of the network structure was captured by the quality of relationships with mother, father, child(ren), spouse/partner, and colleagues. The quality of an individual’s relationship with their parents was also quantified by responses to the following two questions: “Are you satisfied with how you communicate with your mother/father?” and “How close do you feel to your mother/father?”. Answers of “strongly agree” and “quite close” or “agree” and “very close”, respectively, were coded as 1; other answers were coded as 0 [43]. A summed value greater than or equal to 1 was coded as 1, representing a high-quality relationship; other values were coded as 0, indicating low-quality relationships. Similarly, relationship quality with child(ren), spouse/partner, and colleagues was measured with survey questions and coded as dichotomous variables.

**Size** of the network structure was measured by the size of the friendship group and number of siblings. The number of siblings ranged from 0 to 20, with an average of 3. Individuals who had more than three siblings were coded as 1; otherwise, they were coded as 0. The number of friends ranged from 0 to 10, with an average of 3. Individuals with fewer than three friends (the average) were coded as 0, indicating a small friendship network; otherwise, they were coded as 1, representing a large friendship network.

**Accessibility** as a measure of network structure was indicated by an individual’s distance from their parents and whether they were living with their child or children. For access to parents, a participant was coded as 0 if they lived more than 10 miles from their parents and otherwise as 1, with the latter indicating good accessibility [44]. Regarding access to children, an individual was coded as 1 if they lived with their child or children and as 0 if their child or children lived elsewhere. Those who had no children were also coded as 0, indicating the inapplicability of this SN feature. Cronbach’s alpha was calculated for each type of relationship. The values of Cronbach’s alpha were all above 0.70, indicating that the survey instrument had an acceptable level of internal consistency [45].

A logit link function was applied to examine the association between the sociodemographic factors and each latent class identified by LCA. A series of logit regression models were operated to examine whether group classification was associated with the sociodemographic covariates. The dependent variables were indicated by whether the participants belonged to a certain class, with 1 indicating “yes” and 0 indicating “no”. The sociodemographic covariates included gender, ethnic group, income, and education.

In addition, logistic regression and chi-square tests were conducted to further examine Hypotheses 1 and 2. Logistic regression was used to assess the effects of SNs on PY. The dependent variable in logistic regression was indicated by whether the participant was physically active or not. Participants were considered as “physically active” and coded 1 if they performed physical activities five or more times in the last seven days; otherwise, coded as 0, suggesting “physically inactive” [40,41]. Independent variables were all aforementioned SN factors and the sociodemographic factors were included as control variables.

We ran a series of chi-square tests to compare the PA and SN associations across the various socio-demographic groups. The dependent variable in the chi-square tests is *PA*, and the independent variables are the SN indicators above-mentioned, respectively. Taking gender groups as an example, we ran two chi-square tests for the male and female participants separately, to check whether the correlation between the SN indicators and PA varied for the different gender groups. Similar chi-square tests were carried out for the remaining sociodemographic groups.

## 3. Results

Table 2 summarizes the model fit indices for the latent class model. Based on the Akaike information criterion (AIC), Bayesian information criterion (BIC), and Lo–Mendell–Rubin (LMR) likelihood ratio test, the optimal solution was the six-class model. Lower AIC and BIC values indicated better model performance. The LMR likelihood compares the model performance between *n* classes and *n—*1 classes. If the LMR probability is lower than 0.05, the model shows a better fit with *n* classes than with *n—*1 classes [46].

Table 3 summarizes the characteristics of the subgroups classified based on the latent relevance of the participants’ PA and their various network structures. **Class 1** included 21.7% (*n* = 3173) of the total sample. The respondents in Class 1 had a high probability of being physically active (*p* = 0.571). They were more likely to have high-quality relationships with their mothers, fathers, children, and spouses/partners (*p* = 0.976, 0.843, 0.595, 0.737, respectively). They remained in frequent contact with their mothers, fathers, children, and neighbors (*p* = 0.972, 1, 0.717, respectively). Additionally, they were more likely to have larger friend networks (*p* = 0.823) and had easy access to their children (*p* = 0.983) and their spouses/partners (*p* = 0.743).

Approximately 19.7% (*n* = 2728) of the participants belonged to Class 2. The participants in **Class 2** were most likely to be physically active (*p* = 0.599). Regarding various network structures, they were likely to enjoy high-quality relationships with their mothers, fathers, spouses/partners, and colleagues (*p* = 0.875, 0.627, 0.602, 0.901, respectively). Class 2 also had the greatest chance of having high contact frequency with mothers (*p* = 0.968), fathers (*p* = 0.881), and neighbors (*p* = 0.667). They had a high probability of having a large group of friends (*p* = 0.835). However, the participants in Class 2 were less likely to live near their mothers, fathers, and spouses/partners (*p* = 0.195, 0.042, 0.473, respectively). The majority of the participants in this class had no children.

**Class 3** represented 16.4% (*n* = 2393) of the total sample. The respondents in Class 3 were the least likely to be physically active (*p* = 0.473). They tended to have high-quality relationships with their mothers, children, spouses/partners, and colleagues (*p* = 0.593, 0.716, 0.512, 0.879, respectively). They remained in frequent contact with their mothers, fathers, children, and neighbors (*p* = 0.896, 0.797, 0.997, 0.763, respectively). Additionally, they were more likely to have large friend size (*p* = 0.612) and to live with their children and spouses/partners (*p* = 0.599).

**Class 4** represented 14.1% (*n* = 2054) of the participants. The respondents in this class were less likely to be physically active (*p* = 0.499). In terms of social relationships, the participants in Class 4 were likely to have high-quality relationships with their mothers, children, spouses/partners, and colleagues (*p* = 0.924, 0.698, 0.529, 0.891, respectively). They remained in frequent contact with their mothers, fathers, children, and neighbors (*p* = 0.994, 0.854, 0.998, 0.890, respectively). Additionally, these respondents were very likely to live near their mothers (*p* = 1) and to live with their children (*p* = 0.963). Furthermore, they were more likely to have many friends (*p* = 0.657).

About 7.9% (*n* = 1156) of the participants fell into **Class 5**. Class 5 showed a high probability of engagement in PA (*p* = 0.543). The participants in Class 5 were very likely to have no children. Regarding the different dimensions of network structure, these respondents were likely to have high-quality relationships with their mothers, fathers, spouses/partners, and colleagues (*p* = 0.938, 0.892, 0.590, 0.898, respectively). They remained in frequent contact with their mothers, fathers and neighbors (*p* = 0.994, 0.976, 0.837, respectively). They also had a high probability of maintaining friendships with many people (*p* = 0.796). Additionally, they were very likely to live near their mothers and fathers (*p* = 0.950, 1).

**Class 6** included 21.2% (*n* = 3090) of the participants. The respondents in this class were slightly more likely to be physically active (*p* = 0.513) and were more likely to have high-quality relationships with their mothers, fathers, children, spouses/partners, and colleagues (*p* = 0.956, 1, 0.687, 0.641, 0.935, respectively). They had a high probability of being in frequent contact with their mothers, fathers, children, and neighbors (*p* = 0.998, 0.998, 0.999, 0.886, respectively). These participants were also very likely to have many friends (*p* = 0.722) and to live near their mothers, fathers, children and spouses/partners (*p* = 0.947, 1, 0.975, 0.513, respectively).

The results of the logistic regression are summarized in Table 4, illustrating the relationship between each latent class and each sociodemographic factor, namely, gender, ethnic background, income, and education, compared to the other classes. Class 1 had a significant relationship with ethnicity, education, and income. The participants in Class 1 were more likely to be White than Black (*OR* = 0.59, *p* < 0.01), to be well (vs. not well) educated (*OR* = 1.78*, p* < 0.01), and to have a high (vs. a low) income (*OR* = 1.18, *p* < 0.01). Class 2 had significant relationships with gender, education, and income. The participants in Class 2 had a higher probability of being male than being female (*OR* = 0.63*, p* < 0.01), being well- (vs. not well)-educated (*OR* = 2.52, *p* < 0.01), and having a low (vs. a high) income (*OR* = 0.92*, p* < 0.05). Class 3 was significantly associated with all four sociodemographic factors. The participants in Class 3 were more likely to be female than male (*OR* = 1.34, *p* < 0.01), belong to a racial/ethnic minority group than to be White (*OR* = 1.45, *p* < 0.01), to be less well- (vs. well educated) (*OR* = 0.55, *p* < 0.01), and to have a low (vs. a high) income (*OR* = 0.79, *p* < 0.01). Class 4 was also significantly associated with all four sociodemographic factors. The participants in Class 4 were more likely to be female than to be male (*OR* = 1.53*, p* < 0.01), to belong to a racial/ethnic minority group than to be White (*OR* = 1.79*, p* < 0.01), be less well-educated (vs. well-educated) (*OR* = 0.51, *p* < 0.01), and to have a low (vs. a high) income (*OR* = 0.79, *p* < 0.01). Class 5 was significantly associated with gender, ethnic background, and income. The participants in Class 5 were more likely to be male than female (*OR* = 0.66, *p* < 0.01), to be White than to belong to a racial/ethnic minority group (*OR* = 0.84*, p* < 0.05), and to have a high (vs. a low) income (*OR* = 1.36, *p* < 0.01). Class 6 was significantly associated with all four sociodemographic factors. The participants in Class 6 had a higher probability of being female than being male (*OR* = 1.09, *p* < 0.05), being White rather than belonging to an ethnic minority group (*OR* = 0.82, *p* < 0.01), being less well- (vs. well)-educated (*OR* = 0.64, *p* < 0.01), and having a high (vs. a low) income (*OR* = 1.18, *p* < 0.01).

The results of the logistic regression are summarized in Table 5, further demonstrating the effects of SNs on PA (Hypothesis 1). We found that the odds of being physically active increased by 14%, 21%, and 8% if the participants had a good network quality with their father, spouse/partner, and colleagues. The odds rose by 15% for participants who had more friends. Contact frequency with neighbors also increased the individual’s PA engagement by 6%. These are in line with the LCA analysis.

The results of the chi-square tests are summarized in Table 6, further demonstrating that the PA and SN associations varied across different socio-demographic groups. Network quality and the number of friends had significant effects on PA among almost all of the sociodemographic groups. While accessibility and frequency were only significant for some sociodemographic groups. For example, accessibility to and contact frequency with children (*p* < 0.01) were significantly correlated with the female participants’ PA engagements, while contact frequency with neighbors (*p* < 0.01) was significantly related to the male participants’ PA engagements.

## 4. Discussion

The results generally support our first hypothesis. The individual’s SNs, especially network quality and size, had a significant influence on their engagement in PA (Table 3, Table 5, and Table 6). Among the six latent classes, the participants in Classes 1, 2, 5, and 6 were more physically active than those in Classes 3 and 4. Comparing the network structures in these two groups (Classes 1, 2, 5, and 6 vs. Classes 3 and 4), members of the former group were more likely to have high-quality relationships with their mothers, fathers, spouses/partners, and colleagues. The chi-square test and logistic regression results further indicated that the quality of relationships, especially with fathers, spouses/partners, and colleagues, was significantly associated with PA (Table 5 and Table 6). This is generally consistent with the literature, which has shown that high-quality relationships with parents [17], spouses/partners [47], and colleagues [17] provide the social support, resources, and opportunities necessary for people to engage consistently in PA [48].

Having a large network was also found to increase people’s engagement in PA. The physically inactive participants in Classes 3 and 4 were less likely to have large friendship networks, while the physically active participants in Classes 1, 2, 5, and 6 tended to have more friends (Table 3). The chi-square tests and logistic regression results also suggested that the number of friends had a positive association with the individuals’ PA participation (Table 6). However, having more or fewer siblings did not significantly affect the PA engagement for any of the six classes, either physically active or inactive. As previous research has shown, having more friends gives individuals more opportunities to acquire social support, which helps them overcome barriers to engaging in PA [49], whereas sibling homophily is correlated mainly with behaviors that occur at home such as TV watching [50].

In contrast to our expectations, a higher contact frequency and greater accessibility did not show significant positive effects on PA engagement (Table 3 and Table 6). The participants in Classes 1 and 2, who lived far away from their parents, were more physically active than those in Class 4, who lived close to their parents. The logistic regression also found that the odds of being physically active decreased by 21% for participants who lived closer to their father. Similarly, the participants with frequent contact with their neighbors tended to be physically inactive (Classes 3 and 4), while those with less contact with their neighbors were more likely to be physically active (Classes 1 and 2).

Similarly, a previous study found that a high frequency of contact with friends was not a prominent predictor of PA [21]. PA engagement may be determined primarily by interaction with “active” friends, rather than with friends in general [23]. The unexpected findings regarding accessibility and PA may be due in part to the prevalence of virtual SNs. As individuals can interact with others online regardless of spatial distance, the effects of accessibility are diminished.

The analysis supports the second hypothesis. The effects of SNs on the individual’s engagement in PA varied with demographic characteristics such as gender, race, income, and education (Table 4, Table 5 and Table 6).

Gender was found to affect the associations between SNs and PA (Table 4, Table 5 and Table 6). For both the male and female participants, high-quality relationships with fathers and spouses/partners and a large number of friends contributed to increased engagement in PA. However, contact frequency exerted different effects on the engagement of the male and female participants in PA. Males tended to be physically active if they had frequent contact with their neighbors, while female participation in PA was strongly related to their frequency of contact with and access to their children. This is consistent with the claim that having close relationships with neighbors enhances men’s engagement in PA [51], while women spend more time taking care of their children due to their traditional gender roles, which reduce their time to exercise [6].

The effects of SN on PA also differed between White and racial/ethnic minority individuals (Table 4, Table 5 and Table 6). White people tended to be more physically active if they had high-quality relationships with their fathers and spouses/partners, had more friends, had frequent contact with their neighbors, or lived close to their parents and children. Racial/ethnic minority population was more likely to be physically active if they had good relationships with their children and colleagues. Research has demonstrated that White people had larger SNs and received more supportive information and resources than racial/ethnic minority groups [52]. In addition, racial/ethnic minority groups tended to have fewer social connections with their families than White people [53]. Racial/ethnic minority groups rely heavily on informal information obtained from others in their work-related SNs to find jobs [54], which are usually lower-paid jobs [48], leading to more work-related PA [55].

The effects of SNs on engagement in PA varied between participants with different income levels (Table 4, Table 5 and Table 6). High-quality relationships with and proximity to children and spouses/partners significantly promoted engagement in PA for all three income groups. For the high-income participants, high-quality relationships with their mothers and fathers were the main factors increasing their engagement in PA. For the low- and mid-income participants, engagement in PA was significantly associated with access to family, number of friends, and frequency of contact with neighbors. Similarly, the literature has shown that high-income participants tend to have high-quality and supportive SNs, which promote their health-related behaviors (e.g., PA) [56]. Mid- and low-income groups are more likely to have strong connections and frequent contacts with friends and neighbors [57], which help to maintain a cohesive social environment and increase their engagement in PA [32].

Educational attainment also significantly influenced the association between SNs and engagement in PA (Table 4, Table 5 and Table 6). For the participants with low education levels, only high-quality relationships with their mothers and access to their fathers significantly increased their engagement in PA. The moderately and highly educated participants tended to be actively engaged in PA if they had high-quality relationships with their children and spouses/partners or had many friends. Individuals with higher social status tend to have larger SNs and more material resources [29,30,58]. Individuals with higher educational attainment are more likely to adopt health-promoting behaviors exhibited by their social connections [30] as education strongly promotes social support and self-efficacy, which lead to a desirable lifestyle [59].

This study has several limitations. First, due to limitations in the survey data, not all dimensions of SNs were included. For example, we only used the network size to measure the friendship networks but did not include the quality, frequency, and accessibility. Aside from the four frequently used measurements in the literature, network centrality, cohesion, and structural equivalence are also important characteristics of SNs. Due to data limitations, these dimensions were excluded from this study. A primary survey including all of these SN structures would improve our understanding of the effects of SNs on PA. Second, this study examined the effects of SNs on participants in their early adulthood. Longitudinal studies of SNs in childhood, adolescence, and different stages of adulthood are needed to further explore the effects of SNs on PA over time.

In summary, using data extracted from Wave IV of Add Health, this study enriches the literature by examining the influence of multifaceted SN structures (quality, size, frequency, and accessibility) on the engagement of young adults in PA across socio-demographic groups. This research provides practical and policy implications for SN interventions to promote PA. It suggests that when health professionals design and implement SN intervention programs, they should provide services based on the individual’s sociodemographic status. Taking gender difference as an example, to increase the engagement of young men in PA, it is most effective to involve their parents, spouses/partners, and friends. However, for young women, intervention programs should focus on creating environments that enable them to engage in PA, taking into account their children’s safety [60]. Some gyms already provide childcare centers to allow mothers to engage in PA without worrying about their children’s safety. In the workplace, a similar measure is known as the “bring baby to work” plan, which also helps to solve the work–family dilemma for mothers, increases the quality of their spousal relationships, and gives them more opportunities to engage in PA after work.

## Figures and Tables

**Table 1 behavsci-12-00345-t001:** The sample statistics.

(*n* = 14,595)			
	Physically active	>5 times of activities/week	53.55%
	Age	25–34	100.00%
	Gender	Female	53.56%
		Male	46.44%
	Race	Caucasian	70.19%
		Minority	29.81%
	Household income	<30 K	19.78%
		30 K–100 K	64.93%
		>100 K	15.29%
	Education	Less than high	7.64%
		High and equivalent	60.51%
		College and above	31.84%

**Table 2 behavsci-12-00345-t002:** The model-fit indices for the latent class models.

	Number of Classes				
	2	3	4	5	6
Pearson x2	45016.605	36817.718	33517.402	30367.184	27018.362
LR x2	23827.251	17680.871	15252.762	13725.754	12805.317
x2 df	65331	65334	65314	65326	65320
# of parameters	33	50	67	84	101
Log likelihood	−116507.357	−113298.294	−112079.841	−111114.421	−110607.208
AIC	233080.714	226696.589	224293.681	222396.842	221416.417
BIC	233331.130	227076.007	224802.102	223034.265	222182.842
Lo–Mendell–Rubin testing the null hypothesis	1 vs. 2 Classes	2 vs. 3 Classes	3 vs. 4 Classes	4 vs. 5 Classes	5 vs. 6 Classes
LMR probability	<0.001	<0.001	<0.001	<0.001	<0.001

Note: AIC—Akaike information criterion, BIC—Bayesian information criterion.

**Table 3 behavsci-12-00345-t003:** The probabilities of the meeting criteria for six distinct subgroups of all participants.

	Class 1	Class 2	Class 3	Class 4	Class 5	Class 6
	21.70%	18.70%	16.40%	14.10%	7.90%	21.20%
	*n* = 3173	*n* = 2728	*n* = 2393	*n* = 2054	*n* = 1156	*n* = 3090
**PA**	0.571	0.599	0.473	0.499	0.543	0.513
* **Quality** *						
Mother	0.976	0.875	0.593	0.924	0.938	0.956
Father	0.843	0.627	0.471	0.28	0.892	1
Child(ren)	0.595	0.041	0.716	0.698	0.049	0.687
Marriage	0.737	0.602	0.512	0.529	0.59	0.641
Colleagues	0.941	0.901	0.879	0.891	0.898	0.935
* **Size** *						
Siblings	0.207	0.252	0.458	0.384	0.194	0.264
Friends	0.823	0.835	0.612	0.657	0.796	0.722
* **Accessibility** *						
Mother	0.026	0.195	0.06	1	0.95	0.947
Father	0	0.042	0.13	0.251	1	1
Child(ren)	0.983	0.001	0.971	0.963	0.001	0.975
Marriage	0.743	0.473	0.599	0.398	0.317	0.531
* **Frequency** *						
Mother	1	0.968	0.896	0.994	0.994	0.998
Father	0.972	0.881	0.797	0.854	0.976	0.998
Child(ren)	1	0.007	0.997	0.998	0.009	0.999
Neighbors	0.717	0.667	0.763	0.89	0.837	0.886

Note: Item-response probabilities > 0.5 to facilitate interpretation.

**Table 4 behavsci-12-00345-t004:** The odds ratio for predictors on latent class membership (*N* = 14,595).

Covariate	Class 1 (*sig* < 0.01)	Class 2 (*sig* < 0.01)	Class 3 (*sig* < 0.01)	Class 4 (*sig* < 0.01)	Class 5 (*sig* < 0.01)	Class 6 (*sig* < 0.01)
Gender_Female	0.97	0.63 **	1.34 **	1.53 **	0.66 **	1.09 *
Race_Minority	0.59 **	0.98	1.45 **	1.79 **	0.84 *	0.82 **
Education_Higher	1.78 **	2.52 **	0.55 **	0.51 **	1.01	0.64 **
Income_Higher	1.18 **	0.92 *	0.79 **	0.79 **	1.36 **	1.18 **
Intercept	0.11 **	0.06 **	0.45 **	0.25 **	0.11 **	0.59 **

Note: ***Sig*** represents that the logit model is effective at the significance level of 0.01. ***** represents *p* < 0.05; ****** represents *p* < 0.01.

**Table 5 behavsci-12-00345-t005:** The logistic regression for the PA and SN factors in terms of the sociodemographic factors (*N* = 14,595).

	Odds Ratio		95%CI	
* **Quality** *				
Mother	0.98		0.94–1.02	
Father	1.14 *		1.00–1.18	
Child(ren)	1.05		0.99–1.12	
Spouse	1.21 *		1.00–1.25	
Colleagues	1.08 *		1.01–1.15	
* **Size** *				
Siblings	1.00		0.98–1.03	
Friends	1.15 **		1.08–1.22	
* **Accessibility** *				
Mother	1.13		0.92–1.38	
Father	0.79 *		0.65–0.97	
Child(ren)	1.01		0.70–1.48	
Spouse	0.86 *		0.74–1.00	
* **Frequency** *				
Mother	1.02		0.98–1.07	
Father	0.98		0.95–1.02	
Child(ren)	0.99		0.83–1.17	
Neighbors	1.06 *		1.01–1.11	
* **MHHI** *				
30–100 k	1.01		0.86–1.18	
>100 k	1.11		0.88–1.40	
* **Education** *				
High school equ	0.93		0.75–1.16	
>High school	0.92		0.71–1.19	
***Gender_***Female	0.68 **		0.59–0.77	
***Race_***Minority	0.88 *		0.76–1.01	
* **Cons** *	0.23		0.00	
**Pseudo R2**	0.17	$X^{2}$	111.32	*p* < 0.00

Note: ***** represents *p* < 0.05; ****** represents *p* < 0.01.

**Table 6 behavsci-12-00345-t006:** The chi-square results of social networks and physical activity in relation to the socio-demographic factors.

Coefficients	Gender		Race		Income			Education		
	Male	Female	White	Minority	<30 K	30–100 K	>100 K	<High School	High School	≥
										College
* **Mother** *										
Accessibility	309.88 *	246.59 *	346.49 **	266.99	237.08	301.03 *	217.4	227.77	258.15	251.85 *
Frequency	400.28	342.75	365.23	357.59	324.63	386.1	292.1	320.19	398.32	322.08
Quality	262.34	319.48	289.04	263.89	287.24	272.38	332.37 *	387.81 **	287.38	186.88
* **Father** *										
Accessibility	358.76 *	293.49 **	364.28 **	273.22	285.14 **	368.72 **	222.08	266.11 **	319.96 **	263.74 *
Frequency	371.27	396.75	397.89	462.79	328.74	424.34	302.99	302.73	378.77	323.26
Quality	416.68 **	391.13 **	397.240 *	373.31	249.85	319.76	349.37 **	235.32	510.41 **	255.38
* **Children** *										
Accessibility	52.2	84.04 **	80.29 **	60.37 *	55.62 *	65.88 *	42.58	48.66	74.34 **	61.52 **
Frequency	166.43	237.26 **	160.6	228.41	179.75 *	147.13	176.09	109.26	156.16	148.7
Quality	432.57	282.71	439.85	636.34 **	361.65 **	399.51 **	262.21 *	307.24	353.08 *	346.97 **
* **Spouses** *										
Accessibility	110.94 **	36.2	88.26 **	55.37	50.81 *	96.40 **	25.13	43.87	76.74 **	48.35
Quality	725.26 *	532.01 **	59.06 *	36.28	283.13 **	286.89 **	257.16 *	881.14	616.16 *	206.36 **
* **Siblings** *										
Size	1200 **	555.99	816.42	865.04	670.48	968.41 **	804.42 **	555.71	883.91	658.54 *
* **Friends** *										
Size	269.47 **	236.65 **	323.54 **	188.69	181.39 *	304.53 **	149.88	153.46	262.36 **	181.17 *
* **Colleagues** *										
Quality	160.41	154.86	173.69	289.59 **	108.11	176.05	129.02	142.61	210.19 *	112.45
* **Neighbors** *										
Frequency	343.49 **	167.1	319.35 **	272.91	223.83 *	296.79 **	194.63	206.57	272.41 **	205.94

Note: The number are chi-square coefficients. ***** represents *p* < 0.05; ****** represents *p* < 0.01.

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
