# Peer review of "Effects of Social Networks in Promoting Young Adults’ Physical Activity among Different Sociodemographic Groups"

_behavsci, 2022, doi:10.3390/bs12090345_

Round 1
Reviewer 1 Report
Dear Authors, the topic that You convered is of high importance. You emphasise the issue of PA correspondenence for the particular group, what in my opinion is essential.
The structure and logic od Your publication is fortecy. You also use proper references.
The only slight mistakes are connected with sentence construction, some are difficult to understand.
Conluding i recognize Your publication as important and influencial on the area of PA in different groups.
I hope You find my comments useful.
Best regards
Author Response
Thank you for providing us this opportunity to revise our manuscript for reconsideration at the journal. We appreciate the valuable comments and suggestions from the reviewers and have made revisions accordingly. Our detailed responses are listed below after reviewers’ comments. All revisions are under “Track changes” in the manuscript. We hope that the reviewers and editors will find this version of the manuscript improved.
Comment:
The topic that You convered is of high importance. You emphasize the issue of PA correspondence for the particular group, what in my opinion is essential. The structure and logic of Your publication is fortecy. You also use proper references.
The only slight mistakes are connected with sentence construction, some are difficult to understand.
Concluding I recognize Your publication as important and influential on the area of PA in different groups.
I hope You find my comments useful.
Response:
Thank you very much for your comments. We have carefully revised the manuscript and had it proofread by an editorial service (Cambridge Proofreading).
Reviewer 2 Report
The present manuscript is generally well written and organized. The authors address an important and timely research question and the work could have important implications. However, I was confused by the fact that the central hypotheses did not appear to be examined (or examinable) with the data and statistical tests employed. It is possible I misunderstood the authors intent and analysis; if so, I encourage the authors to be clearer. Alternatively, it may be impossible to examine the stated hypotheses in the present work; if so, the manuscript would need to be reframed around different research questions.
More specifically: It is not clear to me how the data and analyses can address hypotheses 1 and 2. Namely, I don’t think the authors are able to establish the ‘influence’ of social networks on physical activity nor how potential ‘effects’ of social networks on physical activity vary as a function of sociodemographic characteristics. What the latent class analysis shows is that when considered together social networks and physical activity level can be best organized into 6 classes and what the logistic regressions show is that class membership can be predicted by sociodemographic characteristics. There is no analysis reported regarding the relation between social networks and physical activity - notwithstanding the fact that the latent class analysis does show that the different classes have 'numerically' different probabilities of being considered high vs low on physical activity and the various social network variables...but to my understanding this is not the same as showing a statistically significant 'effect' or 'influence' of social networks on physical activity.
Given the nature of the data (cross sectional, non-experimental) and the nature of the statistical analyses employed (latent class analysis) it is not possible to conclude that social networks had an “influence” or “effect” on physical activity. Such causal interpretations found throughout the manuscript are unwarranted. Moreover, an inspection of table 3 indicates that the probability of being in the low vs. high physical activity category is largely similar (.47-.60) across the 6 classes identified. The authors have not provided a statistical test of the differences in physical activity for each class.
The exact nature of the chi square tests employed is not clearly explained and the results of the chi square tests are not reported in results section. I found it difficult to understand the information contained in table 5, especially as it pertains to the relation between social networks and physical activity independent of sociodemographic variables (which the authors describe in the discussion section).
Round 2
Reviewer 2 Report
I am satisfied with the revised manuscript. I think the authors have done a good job responding to my initial review and revising the manuscript. I have one remaining small point for the authors and editor to consider. The present work was non-experimental and cross-sectional, therefore, I find causal language such as the 'influence' or 'effect' of SN on PA to be too strong. I would recommend using language such as the ability of SN to 'predict' PA or using words like 'correlation' or 'association' which do not imply causation.